# The Effects of FIFA 11+ Kids Prevention Program on Kinematic Risk Factors for ACL Injury in Preadolescent Female Soccer Players: A Randomized Controlled Trial

**DOI:** 10.3390/children10071206

**Published:** 2023-07-12

**Authors:** Maedeh Taghizadeh Kerman, Claudia Brunetti, Ali Yalfani, Ahmad Ebrahimi Atri, Chiarella Sforza

**Affiliations:** 1Department of Sports Injury and Corrective Exercises, Faculty of Physical Education and Sport Sciences, Bu-Ali Sina University, Hamadan 65167-38695, Iran; maedeh.taghizadeh1988@gmail.com (M.T.K.); yalfani@basu.ac.ir (A.Y.); 2Department of Biomedical Sciences for Health, Università Degli Studi di Milano, 20133 Milan, Italy; claudia.brunetti@unimi.it; 3Department of Sports Physiology, Faculty of Sports Sciences, Ferdowsi University of Mashhad, Mashhad 91779-48974, Iran; atri@um.ac.ir

**Keywords:** anterior cruciate ligament, prevention, landing, kinematics, preadolescent

## Abstract

This study aimed to investigate the effects of the 8-week FIFA 11+ Kids program on kinematic risk factors for ACL injury in preadolescent female soccer players during single-leg drop landing. For this, 36 preadolescent female soccer players (10–12 years old) were randomly allocated to the FIFA 11+ Kids program and control groups (18 players per group). The intervention group performed the FIFA 11+ Kids warm-up program twice per week for 8 weeks, while the control group continued with regular warm-up. Trunk, hip, and knee peak angles (from initial ground contact to peak knee flexion) were collected during the single-leg drop landing using a 3D motion capture system. A repeated measure ANOVA was used to analyze groups over time. Significant group × time interactions were found for the peak knee flexion, with a medium effect size (*p* = 0.05; effect size = 0.11), and peak hip internal rotation angles, with a large effect size (*p* < 0.01; effect size = 0.28). We found that the FIFA 11+ Kids program was effective in improving knee flexion and hip internal rotation, likely resulting in reducing ACL stress during single-leg drop landing in young soccer players.

## 1. Introduction

Anterior cruciate ligament (ACL) tears are known as one of the most serious injuries in soccer players, most of which occur without any contact with other players [1]. The non-contact ACL injury mechanisms often involve changes in direction, pivoting, and landing, executed with lower hip and knee flexion, as well as dynamic knee valgus [2]. This motion pattern intensifies the proximal tibial anterior shear force, increasing the stress on the ligament, potentially leading to lesions [3]. These lesions not only bring financial burdens to the players and clubs, but they have damaging consequences on the player’s career [4,5]. The return-to-sport process after ACL injury may take from 6 to 12 months [6]; however, a return at the same fitness and technical level is not guaranteed. Even after ACL surgery, individuals usually experience patterns of abnormal strength, proprioception, balance, and neuromuscular control, as well as an increased risk of re-injury [7]. The return-to-sport process, and all the related short- and long-term problems, are not different in amateur players; indeed, their rehabilitation may be potentially longer and more complex than that performed in professional athletes due to difficulties in the practical organization of therapy and its costs. Similar problems may arise for adolescents, who may perform insufficient treatment [5,6,7].

The ACL injury rate is higher in females than in males practicing the same sport [8]. The injury risk factors that contribute to this increased risk in females are related to anthropometrics, hormones, neuromuscular ability, and biomechanics [9]. By way of example, females present higher ligament laxity than males and lower hamstring strength with respect to the quadriceps [10]. The ACL injury rate is the highest for adolescent female athletes, rising from around the age of 10 to 12 years old [11]. Although this injury was considered rarer in children than in the adult population, the number of ACL reconstructions is increasing at a higher rate than in adults, due to the higher participation of preadolescents in sport activities [12,13], increasing the number of the investigations on ACL injury biomechanics and preventive program effects in preadolescent populations. It has been identified that prepubertal females (from 8 to 10 years old) demonstrate risky movement patterns during landing activities, commonly characterized by increased knee abduction and decreased knee flexion angles [14], highlighting how the deficits in knee control play a critical role in increasing the ACL injury risk [15]. Knee loading and ACL injury risk can result from increased hip adduction and decreased hip abductor muscle strength during dynamic tasks [15,16].

According to a video analysis study, the trunk position in the sagittal plane during landing may influence lower limb injuries [17]. ACL injuries in basketball female athletes more likely occurred with larger trunk lateral flexion angles [18]. Another factor that contributes to the increase in ACL injury risk among adolescent female athletes may be the inadequate neuromuscular strategy to control the knee in the presence of greater forces during growth [19]. A meta-analysis found that neuromuscular training aimed to reduce knee and specific ACL injury would be most effective in the young females [20]. 

One of these injury prevention programs for soccer is the FIFA 11+ warm-up program, designed by FIFA Medical Assessment and Research Center (F-MARC). Soligard et al., (2008) stated this program can prevent injuries in young female soccer players and reduce the risk of injuries by one-third [21]. Several studies have examined the FIFA 11+ program in 14-year-old players and older and reported a reduction (between 32 and 72%) in lower limb injury incidence [22,23]. In addition, Thompson-Kolesar et al., (2018) reported preadolescent female soccer players participating in the FIFA 11+ program reduced knee abduction angle at initial contact, considered a risk factor for ACL injury, during a jump-landing task [24]. In 2017, F-MARC created the FIFA 11+ Kids program, a new warm-up program tailored for preadolescent subjects aged 7–13 years old, with the purpose of injury prevention [25]. It is believed that preventive programs dedicated to preadolescents could improve the incoming neuromuscular deficits and hormonal changes that increase the risk of ACL injury during adolescence, especially in females. For this purpose, the FIFA 11+ Kids program has been designed to improve attention, coordination, balance, and lower limb and core strength; it can be executed directly on-field without additional equipment [26]. Additionally, other research related to the FIFA 11+ Kids program has shown how to reduce the rate of injuries and treatment costs for both male and female soccer players younger than 14 years old [27]. 

Despite the programs’ benefits for injury prevention, only a few studies have reported changes in kinematic variables after various injury prevention programs in children younger than 13 years old [24,24,28,29]. In particular, it seems that the kid-specific program has not been analyzed so far. Consequently, this study aimed to analyze the effect of the FIFA 11+ Kids program on preadolescent female soccer players. Biomechanical assessments of the effectiveness of ACL injury prevention programs often include the simulation of injury mechanisms in a safe and controlled environment [30]. Trunk and lower extremity kinematic risk factors of ACL injury were thus assessed through single-leg drop landing, considered a valid screening method for ACL injury risk assessment [31]. We hypothesized that there are significant kinematic differences between the intervention and control groups, after 8 weeks of FIFA 11+ Kids program or regular warm-up, respectively. In particular, we expected that the intervention group would show improvements regarding landing kinematic variables commonly associated with ACL injury mechanisms, such as better control of the core, decreased knee abduction, and larger knee flexion.

## 2. Materials and Methods

Forty preadolescent female amateur soccer players were recruited from a local soccer club. The inclusion criteria were healthy participants aged 10 to 12 years, playing soccer two times per week for at least six months. Participants were excluded from the study if they had a history of lower extremity and trunk injury, ACL injury or surgery within the past 6 months, impaired balance, or had previously been involved in another injury prevention program. Before participation, all procedures were explained to each athlete, and informed consent was obtained from parents. The players were randomly assigned to FIFA 11+ Kids (n = 20) and control groups (n = 20). At the end of the pre-test session, the athlete picked up an opaque envelope, including a paper designating the athlete assigned to either the intervention group or the control group. Eventually, during intervention and testing, two players from the FIFA 11+ Kids group and two players from the control group dropped out of the present study; 18 players in each group completed the program (Figure 1). Demographic data are presented in Table 1.

### 2.1. Intervention

The intervention group performed the FIFA 11+ Kids program as a replacement to their regular warm-up, before their training session. Before the beginning of the program, a poster containing the explanation of all exercises along with pictures was presented to the coaches and players [32]. The FIFA 11+ Kids program consists of seven different exercise sections: running, skating jumps, single-leg stance, press-ups, single-leg jumps, spider-man, and sideways roll; each one has five difficulty levels [27]. The program emphasizes unilateral and bilateral balance and coordination, power, lower limb and core muscle strength, and optimization of falling techniques. Exercises were performed in a specified order and sequence. All players started from the first difficulty level. When participants completed the exercise in the identified time and the required number of repetitions, as described by the FIFA 11+ Kids manual provided by the F-MARC [33], they advanced to the next level. To ensure their compliance with the programs, all the training sessions were supervised by the same researcher and all the participants completed levels 1 to 5 at the same time. The FIFA 11+ Kids warm-up program took approximately 20–25 min, allowing everyone to complete all the exercises, which was the normal duration of the regular warm-up included before all practices and games. The control group executed the regular warm-up that included jogging, active stretching, and ball exercises such as dribbling and pass techniques. As suggested in the manual, the FIFA 11+ Kids program should be executed before each practice or game, and 10–12 weeks are recommended to obtain beneficial effects. Due to the COVID-19 pandemic, we had to decrease the frequency of training sessions and, thus, the duration and frequency of the program, with respect to the original instructions provided by FIFA. For this reason, both groups executed the FIFA 11+ Kids or regular program twice per week for 8 weeks.

### 2.2. Data Collection

Pre- and post-test assessment sessions were conducted in the laboratory, collecting kinematic data through an 8-camera motion capture system (Qualisys AB, Goteborg, Sweden) positioned around a calibrated test area, with acquisition frequency of 200 Hz. A total of 45 reflecting markers were placed on the spinous process of C7 and T10 vertebrae, shoulders, scapula inferior angles, sacrum, anterior and posterior superior iliac spines, iliac crests, great trochanters, lateral and medial femoral epicondyles, lateral and medial malleoli, first and fifth metatarsal heads, and heels, based on a Visual 3D marker set. Four clusters with four markers each were placed on the thigh and shank. Participants performed a static trial and, after that, they warmed up on a stationary bicycle for five minutes in the laboratory environment. To perform the single-leg drop landing task, they were instructed to stand on the 30 cm high box on their slightly flexed non-dominant leg, and the dominant leg was kept in front of the box so that the heel was in touch with it. Then, they dropped on the ground with the dominant leg, with a 30 s rest between trials (Figure 2) [31]. 

The dominant leg was that preferably used to kick the ball [27]. To gain familiarity with the testing procedure, the participants practiced the task three times. Three successful trials of the single-leg drop landing were collected. A successful trial was considered when participants landed on one leg with arms crossed in front of their chest and maintained balance for at least three seconds after landing. All participants, independently of the treatment, participated in the pre-test and post-test session. The latter included the same assessment protocol, and it was conducted after 8 weeks, since that was the duration of the prevention program. Soccer players were evaluated by the same blinded assessor for conducting pre-test and post-test sessions.

Peak joint angles were obtained in the interval between initial contact and peak knee flexion during single-leg drop landings. We defined initial contact in the 3D system as the first frame in which ground contact was observed and the markers placed on metatarsal heads reached the lowest vertical position, as the subjects were instructed to land with plantarflexed foot [34]. The markers’ 3D position was recorded using Qualisys Track Manager (v.2.2, Qualisys AB, Gothenburg, Sweden) and then analyzed using Visual 3D (C Motion, Germantown, MD, USA). Data were filtered with a fourth-order zero-lag Butterworth 12 Hz low-pass filter before calculating the dependent variables. Joint angles were calculated using an X–Y–Z Cardan rotation sequence agreement with the antero-posterior, medio-lateral, and vertical axes, respectively [35]. The positive angles indicated knee flexion, knee adduction, knee internal rotation, hip flexion, hip adduction, hip internal rotation, trunk flexion, lateral trunk flexion toward to support leg, and contralateral trunk rotation.

### 2.3. Statistical Analysis

The Shapiro–Wilk and Levene’s tests were used to indicate a normal distribution of the data and homogeneity of the variances, respectively. The independent *t*-test was used to compare the mean difference between the two groups for age, height, and weight variables. A one-way repeated measures ANOVA was used to compare differences between intervention, control groups, and testing time. Time (pre-test and post-test) was the within-group factor and group (intervention and control) was the between-subject factor; their main effects were evaluated, as well as their group x time interaction. The effect sizes using partial eta squared (η^2^) were considered small for 0.01 ≤ η^2^ < 0.06, medium for 0.06 ≤ η^2^ < 0.14, and large for η^2^ ≥ 0.14 [36]. The significance level was set at alpha = 0.05. Statistical analyses were performed using SPSS version 25 (IBM Corporation, Armonk, NY, USA).

## 3. Results

The results of the Shapiro–Wilk and Levene’s tests indicated the normality of the data and homogeneity of the variances of the study (*p* ≥ 0.05). The independent *t*-test indicated that there were no differences between intervention and control group in age (*p* = 0.89), height (*p* = 0.92), and weight (*p* = 0.43).

Significant main time effects were found for the peak knee flexion (*p* = 0.01) and peak hip internal rotation angles (*p* < 0.01), both with large effect sizes. The main group effect was significant only at the peak knee flexion angle (*p* = 0.04), with a medium effect size. Additionally, the significant group × time interaction effects were significant for the peak knee flexion (*p* = 0.05), with a medium effect size, and peak hip internal rotation angles (*p* < 0.01), with a large effect size. The intervention group showed increased peak knee flexion and reduced peak hip internal rotation, with respect to the control group. Descriptive and inferential statistics are reported in Table 2. 

## 4. Discussion

This study aimed to investigate the effect of 8 weeks of the FIFA 11+ Kids program on kinematic risk factors for ACL injury in a group of female preadolescent soccer players, compared to a control group, in single-leg drop landings. We hypothesized that the FIFA 11+ Kids program would have beneficial effects for the intervention group, comparing the pre- and post-test results, with respect to the control group, who performed a regular warm-up.

The most important finding is that intervention group players significantly increased peak knee flexion and reduced peak hip internal rotation, compared to control group players, after the FIFA 11+ Kids program. The current results support the study by Brown et al., (2014) [37]. They discovered female participants exhibited significantly greater peak knee flexion angles following neuromuscular and plyometric training [37]. Landing with increased knee flexion reduces tibia anterior shear force and quadricep activity, decreasing ACL loads [38]. It has been found that about 80% of the ACL injuries in soccer occur with knee extension or early flexion [2], due to the predominant activation of the quadriceps with respect to the hamstrings, favoring the anterior tibial displacement, which is more evident in females [39,40]. A recent study that investigated the effect of a 10-week FIFA 11+ Kids program on isokinetic strength of young soccer players aged 10–12 years old demonstrated that the intervention group improved the strength of knee flexors, which is considered beneficial for ACL injury prevention [41]. 

Furthermore, since excessive hip internal rotation angle is associated with dynamic knee valgus and ACL injury [42], the FIFA 11+ Kids program, that includes gluteal muscle exercises, may have contributed to reducing hip internal rotation angle. This is in line with the study by Pollard et al., who reported that hip angles in the transversal plane significantly decreased during landing after injury prevention training in adolescent female soccer players [43]. Knee kinematics are affected by the hip position, as the knee valgus collapse, that leads to ACL injury, often includes hip internal rotation [10,44].

In contrast to our initial hypotheses, we did not find significant improvement in knee abduction. Knee abduction is one of the most relevant variables during ACL injury risk assessment, as landing with abducted knee may increase knee load and, thus, the chance of incurring injuries [45,46]. In our study, both groups showed a slight reduction of knee abduction angle after 8 weeks of the FIFA 11+ Kids or regular warm-up, so this cannot be considered a benefit related to the execution of the FIFA 11+ Kids program. Another study, investigating the effect of FIFA 11+ program on preadolescent soccer players (10–12 years old), found no significant differences in knee abduction angle during single-leg landings between intervention and control group after 8 weeks of training [30]. 

Based on the trunk kinematic results in the present study, we failed to find significant differences in peak trunk flexion, lateral flexion, and rotation angles between the two groups. In this regard, Jeong et al., (2021) reported that core strengthening exercises improved the trunk and lower extremity muscle recruitment strategies by increasing the trunk flexion angle, vastus medialis/vasus lateralis, and hamstrings/quadriceps activation ratios, which led to increased dynamic stability and reduced the risk of ACL injury [38]. For this reason, the insufficient neuromuscular adaptation in early-pubertal females, related to the rapid increase in height and weight, could elicit an increased stress on the ACL [47]. Nakase et al., (2013) evaluated whole-body muscle activities using Positron Emission Tomography after the second part of FIFA 11+ program in adult males, and they reported that the intervention group demonstrated higher muscle activity for abdominal rectus and hip abductor muscles compared to the control group [48]. Although the FIFA 11+ Kids program is important for injury prevention in preadolescent players, its core training component should be enlarged for young athletes, as they need to improve.

In another study, subjects displayed increased trunk flexion angles during the drop jump test and single-leg squat and increased hip, knee flexion angles, and decreased lateral trunk motion during the single-leg squat after participating in the core muscle training of the FIFA 11+ program for 8 weeks, indicating that core-muscle training improved kinematic variables currently considered as risk factors for ACL injury [49]. The second part of the FIFA 11+ program includes the bench, sideways bench, and Nordic hamstring training, activating the abdominal rectus, gluteus medius, and gluteus minimum muscles; thus, it is rational that the intervention group improved trunk and hip kinematic risk factors. Furthermore, future research should design a modified FIFA 11+ Kids program to additionally target exercises for improvement of the core muscles. Moreover, the correct instructions could be useful for the players to better understand how to control the trunk vertically in the frontal plane during landings.

Despite finding no other kinematic changes, our results reported beneficial effects of the FIFA 11+ Kids warm-up program. Due to the increasing interest in injury prevention, many studies have investigated the effects of the FIFA 11+ Kids and other prevention programs on preadolescent populations, but only a few studies focused on biomechanics. The study by Zareei et al., (2018) evaluated the effects of the 10-week FIFA 11+ Kids on preadolescent soccer players (aged 9 to 16 years old), who performed the program twice per week [50]. They found a significant reduction, and, thus, improvement, in the Landing Error Scoring System (LESS), that is, a clinical assessment of jump-landing biomechanics. In addition to the biomechanical point of view, the FIFA 11+ Kids program has other benefits for preadolescent children. It has been found that an intensive 8-week FIFA 11+ Kids program for children in elementary school has positive effects on physical fitness, such as jumping and running ability, and attentional capacity, including focused, sustained, selective, and divided attention [51]. This result was surprising, not only for the everyday well-being of children, but also because worse cognitive functions are associated with higher risk of sustaining an ACL injury [52]. The study by Rössler et al., (2018) reported that the number of injuries and days lost relative to injuries in children practicing soccer (7–13 years old) was less than half after FIFA 11+ Kids, with respect to the regular warm-up. In addition, as the injury rate decreased, the healthcare costs per player per season were reduced by 59% [53]. These results clearly underline the benefits of a widespread implementation in everyday training, focusing on both professional and amateur children and adolescent players.

There are some limitations in the current study. Only preadolescent female soccer players were included in the present study, and the findings cannot be generalized to males or other athletes. In addition, the limited sample size and the multiple comparisons in statistical analysis might have inflated Type I errors. Moreover, to develop a better understanding of the effect of the FIFA 11+ Kids program, other landing tasks such as double-leg jump and vertical drop jump should be evaluated. In addition, due to the COVID-19 pandemic, the number of participants and the frequency of the practice and game sessions were reduced, and, consequently, so was the duration of the program, with respect to the original instructions provided by FIFA. The FIFA 11+ Kids program should have lasted 10 or 12 weeks, and children should have attended 3–4 sessions per week, which may have reduced the positive effects on kinematic variables. 

## 5. Conclusions

In this study, the 8-week FIFA 11+ Kids program was successful in increasing peak knee flexion and reducing peak hip internal rotation in preadolescent female soccer players, with respect to the control group. Larger knee flexion and lower hip internal rotation have been associated with protective motion patterns for ACL integrity. We found no significant improvement in other kinematic variables associated with ACL injuries, such as knee abduction, hip adduction, and trunk control. In addition, the long-term effectiveness of this study needs to be explored, to further understand the effects of the training in modifying the kinematic risk factors for ACL injury during landing. Nevertheless, we can speculate that the FIFA 11+ Kids program had beneficial effects on landing kinematics. For this reason, we believe that this program should be implemented in everyday training sessions, also considering the additional benefits likely provided by the program. In the literature, there is consensus that preventive programs may be more effective if administered during preadolescence, as this could be considered a proper age target to develop physical abilities, such as strength, balance, jumping, running, and core control. The FIFA 11+ Kids program, or more general preventive programs, could improve deficits that increase during adolescence and expose female adolescents to a high risk of ACL injury.

## Figures and Tables

**Figure 1 children-10-01206-f001:**
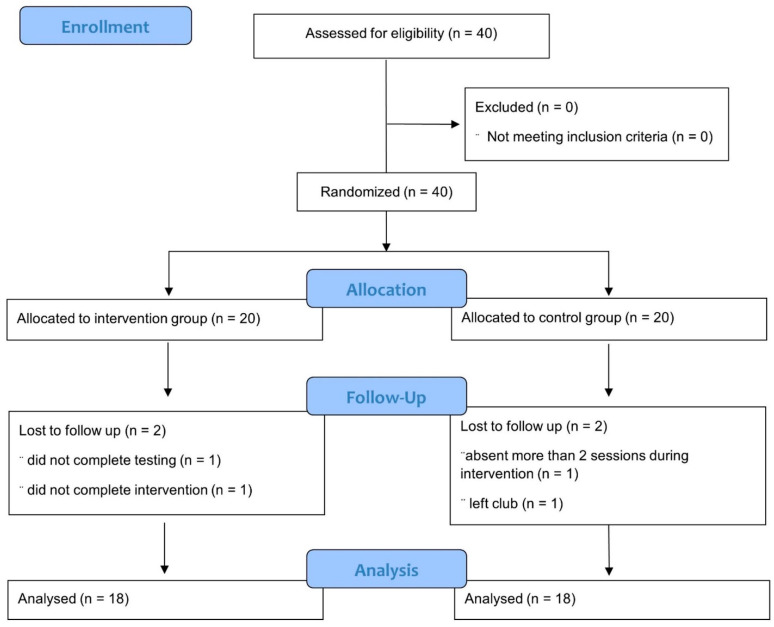
CONSORT flowchart.

**Figure 2 children-10-01206-f002:**
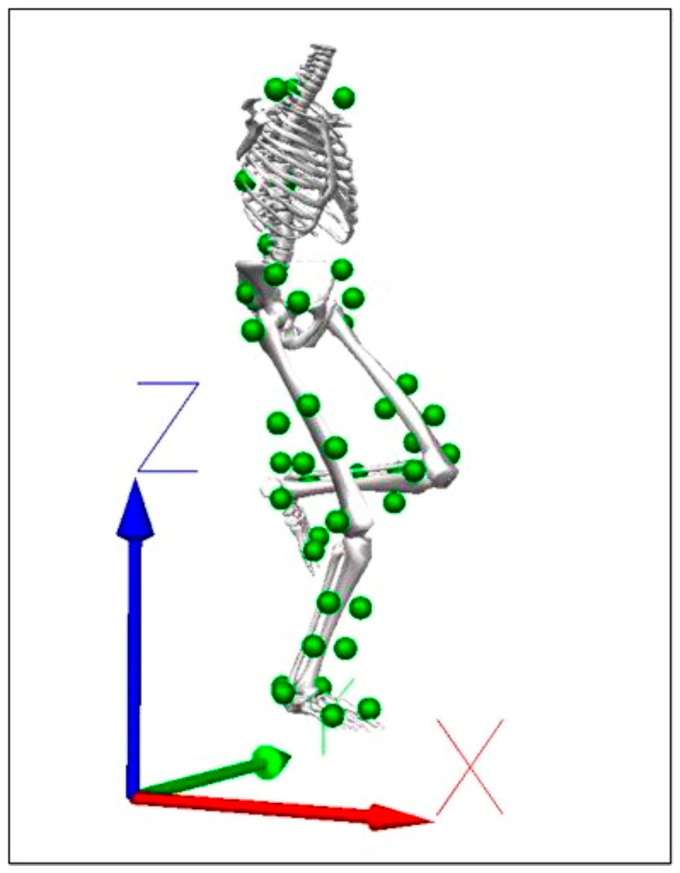
Three-dimensional model of one player while performing a single-leg drop landing task. X: antero-posterior, Y: medio-lateral, and Z: vertical axes. The 45 green spherical dots indicate the position of the anatomical markers (shoulder, trunk, pelvis, lower limb), and of the clustered markers (shank and leg). The Visual 3D software was used for this representation.

**Table 1 children-10-01206-t001:** Demographic characteristics.

	Intervention (n = 18)	Control (n = 18)
Age (y)	11.1 ± 1.2	11.2 ± 1.2
Height (cm)	148.5 ± 5.4	148.3 ± 5.5
Weight (kg)	41.4 ± 3.5	40.5 ± 3.5

Values are means ± SD.

**Table 2 children-10-01206-t002:** Results of one-way repeated measure ANOVA of kinematic variables.

Kinematic Variables (°)	Group	Pre-Test Mean ± SD	Post-Test Mean ± SD	*p*-Value (η^2^)		
				Main Effect of Time	Group × Time Interaction	Main Effect of Group
Peak knee abduction	Control	−2.87 ± 5.11	−2.20 ± 4.40	0.09 (0.08)	0.41 (0.02)	0.66 (0.01)
	Intervention	−3.73 ± 3.81	−2.54 ± 2.93			
Peak knee flexion	Control	32.04 ± 7.19	32.87 ± 6.59	0.01 * (0.19)	0.05 * (0.11)	0.04 * (0.12)
	Intervention	33.36 ± 10.89	40.63 ± 5.35			
Peak knee rotation	Control	−0.32 ± 6.08	0.59 ± 6.45	0.19 (0.07)	0.47 (0.01)	0.91 (0.00)
	Intervention	−0.90 ± 8.21	0.58 ± 8.71			
Peak hip adduction	Control	6.23 ± 3.43	5.86 ± 3.84	0.26 (0.03)	0.29 (0.03)	0.77 (0.00)
	Intervention	6.04 ± 5.49	6.89 ± 5.60			
Peak hip flexion	Control	14.13 ± 11.37	13.87 ± 10.24	0.16 (0.06)	0.24 (0.04)	0.18 (0.05)
	Intervention	17.15 ± 9.17	19.60 ± 9.30			
Peak hip internal rotation	Control	11.48± 14.51	15.88 ± 10.50	<0.01 * (0.31)	<0.01 * (0.28)	0.86 (0.00)
	Intervention	16.18 ± 8.79	13.08 ± 8.32			
Peak trunk lateral flexion	Control	−12.37 ± 6.06	−12.24 ± 4.55	0.64 (0.01)	0.56 (0.01)	0.97 (0.00)
	Intervention	−12.04 ± 3.49	−11.80± 5.35			
Peak trunk flexion	Control	20.58 ± 9.21	21.99 ± 7.83	0.46 (0.01)	0.31 (0.03)	0.12 (0.07)
	Intervention	17.32 ± 7.06	17.10 ± 7.96			
Peak trunk rotation	Control	−8.47 ± 6.34	−8.37 ± 6.65	0.22 (0.04)	0.35 (0.03)	0.39 (0.02)
	Intervention	−11.26 ± 8.91	−9.73 ± 7.69			

* Indicates statistical significance *p* < 0.05.

## Data Availability

The data presented in this study are openly available in Mendeley Data at doi:10.17632/2bmpyfydn4.2. https://data.mendeley.com/datasets/2bmpyfydn4/3 (accessed on 30 June 2023).

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
