# Peer review of "The Effects of FIFA 11+ Kids Prevention Program on Kinematic Risk Factors for ACL Injury in Preadolescent Female Soccer Players: A Randomized Controlled Trial"

_children, 2023, doi:10.3390/children10071206_

Round 1

Reviewer 1 Report

Dear authors,

Thank you for the opportunity of reviewing your manuscript. Overall, the topic is interesting, but the paper presents serious flaws that should be addressed before considering it for publication.

Introduction

1) The authors stated that the study was developed in preadolescent female soccer players aged between 10 and 12. However, the WHO says adolescence is between 10 and 18 years, and there are previous reports that girls' adolescence period may range between 8 and 19. Therefore, I disagree with the term “preadolescent female soccer players”. Besides, between 10 and 12 years, there is a high possibility of menarche occurrence, which is a strong indicator of biological maturation. Overall, the effects of growth and biological maturation on girls’ physical development should be exploited in this section. The authors should consult the following book:

Malina, R. M., Bouchard, C., & Bar-Or, O. (2004). Growth, maturation, and physical activity. Human kinetics.

2) Hyphotesis: “We hypothesized that there are significant kinematic differences between the intervention and control groups, after 8 weeks of 11+ Kids program or regular training, respectively.” This is vague. Differences favoring which group?

Materials & Methods

3) During the text, the authors often use “soccer,” but it is also referred to as “football”. Please make sure that there is consistency in the term used. 

4) Please refer to and provide the Ethics Committee approval in the text.

5) please ensure the numbers below 10 are entirely written in the text. 

6) “Participants performed a static trial and after that, they warmed up on a stationary bicycle for 5 minutes in the laboratory environment. To get familiar with the testing procedure, the participants practiced the task three times.” It is not clear for the reader which task they performed. The authors should refer to the single-leg drop jump first, explain the task, and then explain the details concerning the testing phase. 

7) Please explain who performed the data collection; where it was performed; when it was performed. Please also explain that pre and post-intervention were performed 8 weeks apart since this is unclear. 

8) Data analysis (SPSS) should be presented in a separate data processing section. 

9) Table 1 should be presented in the Results section since it shows descriptive characteristics and the results of the comparison between groups. 

Results

10) Overall, the results are mostly descriptive. Please include, at least, the p-value and effect size for each analysis. 

11) Please check carefully Table 2 and correct some presentation errors (e.g. peak hip flexion and peak knee flexion results).

Discussion/Conclusion

12) Please include the implications of the results for the training process. 

Reviewer 2 Report

Title

In some studies this program is called FIFA 11+Kids, check the official name of the program.

The intervention group performed the 11+ Kids program replacing their regular training.

The FIFA 11+ kids program is not a substitute for training.

Data collection

Before the beginning of the program, a poster containing the explanation of all exercises along with

pictures was presented to the coaches and players (https://physiofitadelaide.com.au/fifa-

11-for-kids/).

On this website the program is called: The FIFA 11+ for Kids! A little lower on the page there is also this term: The 11+ Kids Warm Up.

A total of 45 reflecting markers were placed on the spinous process of C7 and T10 vertebrae, shoulders, scapula inferior angles, sacrum, anterior and posterior superior iliac spines, iliac crests, great trochanters, lateral and medial femoral epicondyles, lateral and medial malleoli, and first and fifth metatarsal heads, heels, based on a Visual 3D marker set.

Are the places where the markers are placed part of the standard procedure, if so add a reference.

The CG executed their regular training.

Regular training is also mentioned here, I think it would be correct: The CG executed their regular warm-up.

What does a regular warm-up consist of, write in a few sentences (eg, running, stretching exercises...).

Both training took approximately 20-25 minutes, which was normally included before all practices and games, twice per week for 8 weeks.

From this sentence we can guess that it is about warming up, but it should be clearer. From the same sentence we can conclude that the whole training takes 20-25 minutes, which is wrong. You can add information about the total duration of the training.
In some studies, this program lasts about 15 minutes. It is necessary to comment on the duration of the treatment, because it is obviously not standardized.
This chapter needs to be worded more clearly.
There is no chapter on sample variables, it should be added.

Data processing

The first part of this chapter is more about measurement methodology, the second part is about data processing methods. They can be separated, then it would be clearer.
The discussion should be extended, much has been written in science about this program, and some parallels can be drawn with other papers.

Study limitations

In addition, due to the COVID-19 pandemic, we had to decrease the frequency of the sessions and the total duration of the program, with respect to the original instructions provided by FIFA. The 11+ Kids programshould have lasted 10 or 12 weeks, and children should attend 3-4 sessions per week,

This should be written in the chapter experimental procedure. Until the end of the article, it was not clear to me why the trainings are only 2x a week. And the conclusion can be longer.

Round 2

Reviewer 1 Report

Dear authors,

I appreciate your effort in improving the manuscript.

I accept the manuscript in its current form.

Reviewer 2 Report

Paper can be published.

Best regards,

Hrvoje